# Effects of a Single Ingestion of Trehalose during Prolonged Exercise

**DOI:** 10.3390/sports7050100

**Published:** 2019-04-29

**Authors:** Tsuyoshi Wadazumi, Kanji Watanabe, Hitoshi Watanabe, Hisayo Yokoyama, Nobuko Hongu, Norie Arai

**Affiliations:** 1Faculty of Health and Well-being, Kansai University, Sakai-shi 590-8515, Japan; 2School of Health and Sports Sciences, Mukogawa Womes’s University, Nishinomiya-shi 663-8558, Japan; kanji_w@mukogawa-u.ac.jp; 3Research Center for Urban Health and Sports, Osaka City University, Osaka-shi 558-8585, Japan; watanabe@sports.osaka-cu.ac.jp (H.W.); yokoyama@sports.osaka-cu.ac.jp (H.Y.); 4Department of Nutritional Sciences, The University of Arizona, Tucson, AZ 85721, USA; hongu@email.arizona.edu; 5R&D Division, Hayashibara Co., Ltd., Okayama-shi 702-8006, Japan; norie.arai@hb.nagase.co.jp

**Keywords:** trehalose, dietary supplement, nutrition, exercise performance, lipid utilization

## Abstract

Trehalose (TRE), a disaccharide, is absorbed slowly and gradually increases the blood glucose (GLU) level along with reducing insulin secretion. The aim of this study was twofold. First, we examined exercise performance following ingestions of either GLU, TRE, or water (WAT). The second purpose was to investigate the effects of TRE energy metabolism during prolonged exercise. We examined exercise performance using the Wingate test, with 30-min constant load exercise at 40% VO_2_peak after exercising for 60 min at 40% VO_2_peak, by using an electromagnetic brake-type bicycle ergometer (Part 1). The power values, blood glucose and lactate, and respiratory exchange ratio (RER) were measured. In addition, we investigated the energy metabolism after a single ingestion of TRE, by measuring the RER and estimating the lipid oxidation for 60 min at 40% VO_2_peak (Part 2). Healthy college male students performed three trials—(1) placebo (WAT), (2) GLU, and (3) TRE. Repeated two-way analysis of variance (ANOVA) was used for a comparison of the data among the three trial groups. A multiple comparison test was performed using post hoc Bonferroni correction. The TRE ingestion significantly increased the average and maximum power values (*p* < 0.01). The TRE ingestion showed significantly higher lipid utilization than the GLU lipid oxidation values the in TRE, 12.5 ± 6.1 g/h; GLU, 9.3 ± 4.7 g/h; and WAT, 15.0 ± 4.4 g/h; (*p* < 0.01). In conclusion, we provide novel data that a single TRE ingestion was effective in improving prolonged exercise performance by effective use of glucose and lipids.

## 1. Introduction

Glycogen stored in the body is used as a source of energy during exercise. During the later stages of exercise in prolonged sports events, such as a full marathon, muscle glycogen is continuously consumed, and the depletion of glycogen causes a decrease in performance [1,2,3]. The later stages of prolonged exercise cause dramatic reductions in the muscle glycogen content in skeletal muscle [4]. However, in such prolonged endurance exercise, the ingestion of carbohydrates is shown to delay fatigue onset and prolong exercise duration [5]. In addition, the storage and sparing of muscle glycogen [6,7] and the maintenance of higher blood glucose levels are also useful for maintaining or improving performance in such exercise [8,9]. Thus, increasing glycogen stores and maintaining higher glucose levels by carbohydrate ingestion are important factors for improving performance in endurance exercise.

The representative method for increasing glycogen stores is called “glycogen loading”. In this method, muscle glycogen re-synthesis is enhanced by depleting the muscle glycogen stores by heavy physical activity, and then eating a diet rich in carbohydrates (exhaustive depletion followed by a carbohydrate-rich diet) [10]. Other methods include the ingestion of carbohydrate drinks, such as sports drinks for nutritional support before or during exercise. Many preceding studies have investigated the relationship between the effect of carbohydrate ingestion in improving performance and the types of carbohydrates ingested during exercise, as well as their ingestion method [11,12,13,14,15,16,17]. However, because absorption properties and availability in skeletal muscles differ depending on the types of carbohydrates ingested before or during exercise [18,19], foundational data regarding nutrient dosing and timing, which may enhance exercise performance, has yet to be established for trehalose (TRE).

Glucose, which is a carbohydrate classified as a monosaccharide, causes a rapid increase in blood glucose levels after being ingested, because it is absorbed by the small intestine, and then directly delivered to the blood. The ingestion of carbohydrates, including glucose, inducing a rapid increase in blood glucose levels immediately before prolonged exercise is associated with the additional secretion of insulin following hyperglycemia; this may cause hypoglycemia during the early stages of exercise, resulting in a decrease in performance [20,21]. For this reason, some studies concluded that glucose is not appropriate as a carbohydrate to be ingested for prolonged exercise [21]. Recently, the utilization of TRE, which has become inexpensive and easily available, has attracted attention in various fields. TRE, a hydrate in which two D-glucose molecules are bound to each other, is absorbed slowly by the small intestine, because it requires the enzyme trehalase for its digestion. The activity of trehalase varies among individuals; blood glucose levels are less likely to increase after ingestion by individuals with a poor activity of trehalase [22]. In addition, TRE, which is absorbed more slowly than GLU and other monosaccharides and other disaccharides, generally induces only a slight insulin response after being ingested [23,24]; therefore, hypoglycemia is less likely to occur, even after the ingestion of a large amount of TRE. Furthermore, TRE, which is absorbed slowly and continuously, supplies GLU to the blood, and might allow for the maintenance of higher blood glucose levels during the later stages of prolonged exercise.

It is reported that after the ingestion of GLU or sucrose, lipid oxidation is inhibited by a rapid increase in blood glucose levels, whereas carbohydrate oxidation increases [25]. In contrast, as mentioned above, TRE causes no such rapid increase in blood glucose levels, and induces only a slight insulin response. TRE takes longer to digest than most sugars, therefore, after the ingestion of TRE, lipid oxidation may be less likely to be inhibited, leading to the preservation of carbohydrates; this indicates that TRE is suitable as a carbohydrate to be ingested before prolonged exercise. The utilization of such physiological properties of TRE could maintain higher blood glucose levels than the ingestion of GLC for a long period of time, leading to the maintenance or improvement of performance during the later stages of prolonged exercise. However, only a few studies have investigated the relationship between TRE ingestion and exercise performance from such a perspective. Jentjens and Jeukendrup [26] examined the effects of pre-exercise ingestion of TRE on metabolic responses at rest and during exercise, and on subsequent time-trial performance. They reported the ingestion of TRE leads to lower glucose and insulin responses prior to exercise, and reduces the prevalence of rebound hypoglycemia compared with the ingestion of glucose. However, TRE ingested before exercise did not affect time-trial performance [26]. Thus, the aim of this study was to investigate the effects of a single ingestion of TRE on exercise performance (Part 1) and energy metabolism (Part 2) during prolonged exercise. We hypothesized that a single ingestion of TRE solution would promote greater performance benefits because of the higher lipid oxidation (remaining significantly lower respiratory exchange ratio (RER)) and maintaining higher blood glucose levels in the TRE than GLU trial.

## 2. Materials and Methods

### 2.1. Participants

Healthy recreationally trained college male students were enrolled in this study. This study was conducted in accordance with the Declaration of Helsinki, and approved by the Ethics Committee of the Faculty of Health and Well-being, Kansai University (2016-4). All of the participants received oral information about the method and possible risks of this study, and voluntarily signed the consent form after understanding the contents thereof. In Part 1, a total of 25 healthy male college students participated in this experiment (age 21.3 ± 1.1 years; height 172.7 ± 5.0 cm; weight 66.4 ± 6.9 kg). The participants in Part 2 consisted of a total of 24 males college students (age 21.9 ± 1.2 years; height 173.4 ± 5.6 cm; weight 66.2 ± 5.8 kg). None of the participants in Group 1 were included in the Part 2 experiment. Table 1 and Table 2 show the physical characteristics of the participants. In both experiments (Part 1 and Part 2), a total of three trials—(1) placebo t (water—WAT), (2) GLU trial, and (3) TRE—were performed, one week apart. The participants were instructed to maintain their normal diet and physical activity during the study period. In addition, they were not allowed to perform vigorous physical activity, ingest caffeine, or drink alcohol from 24 h before each trial, and they fasted from 21:00 on the day before each trial, to make it easier to produce a depletion of liver and muscle glycogen. All of the examinations started from 09:00. The participants completed all of the trials and there were no drop-outs in the study.

### 2.2. Experimental Design—Part 1 and Part 2

The experiments were randomized and had repeated-measures, in which the participants reported for trials in Part 1 or Part 2. Part 1 examined exercise performance, comparing the GLU and TRE ingestions, and Part 2 investigated the benefits of TRE regarding its energy metabolism. Figure 1 and Figure 2 show an overview of the experimental designs, respectively. Upon the participants’ first visit to the lab, their height and body mass were recorded. The participants practiced the Wingate test in advance, and performed the main trials after becoming familiar with the test. Prior to the main trials, the VO_2_peak (i.e., the highest value of VO_2_ attained during maximum physical effort, designed to bring a participant to the limit of tolerance) was identified for each subject using an expired-gas analyzer (Aero Monitor AE-310S; Minato Medical Science, Tokyo, Japan), and a 40% VO_2_peak and exercise load (watts) corresponding to a 40% VO_2_peak were determined by using the obtained VO_2_peak value. Treha^®^ (Hayashibara, Okayama, Japan) and glucose (Wako Pure Chemical Industries, Osaka, Japan) were used for testing TRE and GLU, respectively. The TRE or GLU solutions were prepared at a concentration of 8%. Water was used as a control.

#### 2.2.1. Part 1: Exercise Performance Protocol

The exercise performance was assessed by using ultra-high-intensity intermittent exercise (the Wingate test). The Wingate test was performed by using an electromagnetic brake-type bicycle ergometer (PowerMax VIII; Combi Wellness, Tokyo, Japan), the pedals of which were given a load of 0.075 kg/kg body weight [27]. Each set of exercise consisted of three bouts of 30-s full-power cycling, with a 4-min recovery between each bout (see Figure 1—Wingate test).

After fasting for 12 h, the participants performed a constant-load exercise at 40% VO_2_peak for 60 min by using an electromagnetic brake-type bicycle ergometer. Subsequently, they performed four sets of the Wingate test, between 30-min of constant-load exercise at 40% VO_2_peak cycling. The procedure of this experiment was designed to develop a physiological stage when runners start rely on fat for fuel, and start slowing down, known as hitting “runner’s wall” in a full marathon or endurance running events. The participants performed the examination of the same protocol a total of three times by randomly changing the type of test drink to be ingested. The test drink ingested at the end of the first set was WAT, or an 8% GLU or TRE solution. The participants ingested 500 mL of the test drink within 5 min after the end of the first set (Figure 1).

#### 2.2.2. Part 2: Energy Metabolism Protocol

The energy metabolism during the low-intensity constant-load exercise was assessed by measuring the RER. Figure 2 shows the protocol of exercise. After a 5-min rest period, the subjects performed constant-load exercise at a 40% VO_2_peak for 30 min, by using an electromagnetic brake-type bicycle ergometer. After the end of the 30-min constant-load exercise at a 40% VO_2_peak, they ingested the test drink, and then performed the same constant-load exercise for another 60 min. The subjects ingested 500 mL of the test drink within 5 min for each trial (Figure 2). The subjects performed the examination of the same protocol a total of three times by randomly changing the test drink to be ingested. The examinations were performed at least one week apart.

### 2.3. Assessments—Exercise Performance (Part 1) and Energy Metbolism (Part 2)

In Part 1, the exercise performance was assessed using the Wingate test. The results of each Wingate test were expressed as (1) the average power values (Figure 3a) and (2) the maximum power values (Figure 3b). The average power values (Watts) were determined for each 5-s period for 30 s, and they (5 s × six readings = 30 s) were averaged as a bout of the Wingate test. The participants were instructed to maintain a maximal pedal speed throughout the 30 s period. The maximum power value (Watts) was the highest power produced in a 30-s segment. The 30-s segment of the test were repeated three times between 4 min of resting period; this is one bout of Wingate test (see Figure 1—Wingate test). The participants were motivated verbally throughout the test.

The changes in both of the indicators are presented by the data (Watts) and data expressed percentages (%) of that in the first set, as 100%. The glucose and lactate levels in the blood collected from the fingertips were measured by using a small blood glucose meter (Glutest Neo Super; Sanwa Kagaku Kenkyusho, Nagoya, Japan) and a simplified blood lactate test meter (Lactate Pro2; ARKRAY, Kyoto, Japan), respectively. The blood glucose and lactate levels were measured in duplicates 12 times, between at rest, immediately before the examination, and the end of the examination (① to ⑫ in Figure 1). When the differences of these two measurements were more than 10%, we took the third measurement. The RER was also determined concurrently with these parameters. In addition, various ventilation rates were determined a total of five times for the 5 min at rest, as well as for the last 15 min of each constant-load exercise (I–V in Figure 1), by using an expired-gas analyzer (Aero Monitor AE-310S; Minato Medical Science, Osaka, Japan).

In Part 2, the blood glucose levels were measured at the following six time-points for each examination: at rest, immediately before the examination; at the end of the 30-min constant-load exercise; and every 15 min after the start of the 60-min constant-load exercise (① to ⑥ in Figure 2). In addition, various ventilation rates were determined continuously (① to ⑥ in Figure 2, except while the participants were ingesting 500 mL of the test drink, within 5 min) using an expired-gas analyzer (Aero Monitor AE-310S; Minato Medical Science, Osaka, Japan). The substrate oxidation during exercise was calculated using the following formula, according to a preceding study [28].
Lipid oxidation (mg/min) = 1.689 × (VO_2_ − VCO_2_) − 1.943 × Nu(1)

### 2.4. Statistical Analysis

The obtained data are expressed as the mean ± standard deviations (SD). A two-way (solution × time) repeated measures ANOVA was used for the comparison of the data among the three trial groups. When a significant difference was detected, a multiple comparison test was performed using the post hoc Bonferroni correction. All of the analyses were performed using SPSS version 25.0 (IBM Japan, Tokyo, Japan), and the statistical significance was set at *p* < 0.05.

## 3. Results

### 3.1. Assessment of Exercise Performance—Part 1

Figure 3a, the measured value, and Figure 3b, the % value, show the results of the comparison of the average power value among the exercise sets for each trial (ingestion of TRE, GLU, or WAT solution). The performance levels were the highest in the first set, and then decreased with increasing the number of sets in all three of the trials in both the measured (Figure 3a) and % (Figure 3b) values. In the third set, the performance levels were significantly higher in the TRE (95.8 ± 4.7%) and GLU (95.1 ± 4.3%) trials than in the WAT trial (91.9 ± 4.8%) (WAT vs. GLU, *p* < 0.001; WAT vs. TRE, *p* < 0.001). In the fourth set, the performance levels were significantly higher in the TRE trial (93.1 ± 5.4%) than in the GLU trial (90.0 ± 5.4%) and WAT trial (86.4 ± 7.7%) (WAT vs. TRE, *p* < 0.001; GLU vs. TRE, *p* < 0.01). The measured average power value of the TRE in the fourth set was also significantly higher than the GLU and WAT (Figure 3a). This is because, in the TRE trial, the decrease in the average power value from the third set was milder than in the GLU trial and WAT trial, leading to the maintenance of the performance levels. Figure 4a, the measured value, and Figure 4b, the % value, show the results of the comparison of the maximum power value among the exercise sets for each trial. The performance levels were higher in the second set than in the first set in all three of the trials. In the fourth set, the TRE trial achieved a high-performance level; the performance levels were significantly higher for the TRE trial (102.9 ± 5.4%) than for the WAT trial (98.3 ± 7.6%) and GLU trial (100.3 ± 3.6%) (WAT vs. TRE, *p* < 0.01; GLU vs. TRE, *p* < 0.01). There were no significant differences in the measured values in the maximum power value among the exercise (1–4 sets) for each trial (WAT, GLU, or TRE; Figure 4b).

### 3.2. Changes in Blood Glucose and Lactate Levels—Part 1

Figure 5 shows the changes in the blood glucose levels for each trial. The blood glucose levels increased sharply after each set of the Wingate test in all three of the trials (WAT, GLU, and TRE). After the ingestion of the test drink (WAT, GLU, or TRE), each trial exhibited changes in the blood glucose levels. At the time-point immediately after the test-drink ingestion (④ in Figure 5), the blood GLU levels did not differ greatly among the three trials. However, 15 min after the test-drink ingestion (⑤ in Figure 5), the blood glucose levels increased mildly in the TRE trial (95.5 ± 18.0 mg/dL), increased sharply in the GLU trial (110.4 ± 18.7 mg/dL), and decreased slightly in the WAT trial (81.0 ± 11.4 mg/dL); the three trials exhibited significantly different blood glucose levels from each other because of the effects of carbohydrate ingestion (WAT vs. GLU, *p* < 0.001; WAT vs. TRE, *p* < 0.01; GLU vs. TRE, *p* < 0.01). At the end of the second set (⑥ in Figure 5), the blood glucose levels were comparable for the TRE trial (113.8 ± 26.0 mg/dL) and the GLU trial (113.5 ± 20.1 mg/dL), and were significantly higher in both the TRE and GLU trials than in the WAT trial (94.4 ± 11.9 mg/dL) (WAT vs. GLU, *p* < 0.001; WAT vs. TRE, *p* < 0.01). From the end of the third set (⑨ to ⑫ in Figure 5), the blood glucose levels remained significantly higher in the TRE and GLU trials than in the WAT trial. From the time-point of ⑩ in Figure 5, the blood glucose levels in the WAT trial began to decrease below those at rest. At the time-point of ⑫ in Figure 5, the blood GLU levels were significantly higher in the TRE trial (86.4 ± 15.1 mg/dL) than in the WAT trial (77.1 ± 13.2 mg/dL) (WAT vs. TRE, *p* < 0.05).

Figure 6 shows the changes in the blood lactate levels for each trial. In all three of the trials (WAT, GLU, and TRE), the blood lactate levels increased after each set of the Wingate test, and decreased during the subsequent constant-load exercise. At the time-point of ⑫ in Figure 6, the changes in the blood lactate levels were significantly higher in the TRE trial (16.9 ± 1.1 mmol/L) than in the WAT trial (12.7 ± 1.0 mmol/L) (WAT vs. TRE; *p* < 0.05).

### 3.3. Changes in RER—Part 1

Figure 7 shows the changes in RER for each trial. After the ingestion of the test drink, the trial GLU exhibits higher RER levels until the final phase of the examination, in contrast to the other two trials. At 15 min after test-drink ingestion (III in Figure 7), the RER was significantly higher in the GLU trial (0.89 ± 0.04) than in the TRE trial (0.86 ± 0.03) (GLU vs. TRE, *p* < 0.05). At the time-point of IV in Figure 7, the RER was significantly higher in the GLU trial (0.88 ± 0.03) than in the trials of TRE (0.85 ± 0.03) and WAT (0.83 ± 0.03) (WAT vs. GLU, *p* < 0.001; GLU vs. TRE, *p* < 0.05).

The RER was determined at the time-points of I to V in the exercise protocol, by using an expired-gas analyzer to observe the changes in the parameter.

### 3.4. Assessment of Energy Metabolism in Low-Intensity Constant-Load Exercise—Part 2

Figure 8 shows the changes in the blood glucose levels for each trial in Part 2. After the ingestion of the test drink, each trial exhibited changes in the blood glucose levels, reflecting the properties of its test drink. After 15 min of the test-drink ingestion (③ in Figure 8), the blood glucose level was increased mildly in the TRE trial (88.6 ± 2.2 mg/dL), increased sharply in the GLU trial (105.3 ± 2.2 mg/dL), and decreased slightly in the WAT trial (80.5 ± 1.3 mg/dL); the three trials exhibited significantly different blood glucose levels from each other because of the effects of the carbohydrate ingestion (WAT vs. TRE, *p* < 0.05; WAT vs. GLU, *p* < 0.001; GLU vs. TRE, *p* < 0.001). Between 30 and 60 min (at the end of the trial) after test-drink ingestion (④ to ⑥ in Figure 8), the blood glucose levels were maintained in the TRE trial (89.0 ± 1.7 to 84.8 ± 1.2 mg/dL), decreased greatly in the GLU trial (99.2 ± 2.5 to 80.9 ± 1.4 mg/dL), and decreased more greatly in the WAT trial (78.8 ± 1.1 to 75.8 ± 1.3 mg/dL). After 60 min of the test-drink ingestion (⑥ in Figure 8), the blood glucose levels were significantly higher in the trials of the TRE (84.8 ± 1.2 mg/dL) and GLU (80.9 ± 1.4 mg/dL) than in the WAT trial (75.8 ± 1.3 mg/dL) (WAT vs. TRE, *p* < 0.001). The blood glucose levels were always significantly lower in the WAT trial than in the other two solutions between 15 and 60 min after the test-drink ingestion.

Figure 9 shows the changes in the RER for each trial. Between 30 and 60 min (at the end of the trial) after the test-drink ingestion (⑦ to ⑩ in Figure 9), the RER remained significantly higher in the GLU trial (0.97 ± 0.04 to 0.96 ± 0.04) than in the trials of TRE (0.95 ± 0.05 to 0.94 ± 0.04) and WAT (0.93 ± 0.04 to 0.91 ± 0.04) (⑦: GLU vs. TRE, *p* < 0.05; WAT vs. GLU, *p* < 0.001 to ⑩; GLU vs. TRE, *p* < 0.01; WAT vs. GLU, *p* < 0.001). Between 40 and 60 min (at the end of the trial) after the test-drink ingestion (⑧ to ⑩ in Figure 9), the RER decreased slightly in the TRE trial, to the later stages of exercise (0.95 ± 0.05 to 0.94 ± 0.04), whereas, it decreased markedly in the WAT trial (0.92 ± 0.04 to 0.91 ± 0.04). The RER was significantly higher in the TRE trial than in the WAT trial (⑧: TRE vs. WAT, *p* < 0.01, to ⑩: TRE vs. WAT, *p* < 0.01). After 40 min of the test-drink ingestion, the three trials exhibited significantly different RER values from each other.

Figure 10 shows the results of the comparison of the total lipid oxidation (L-value) during the 60-min exercise among the three trials. The L-values were significantly higher in the TRE (12.5 ± 6.1 g/h) and WAT (15.0 ± 4.4 g/h) trials than in the GLU trial (9.3 ± 4.7 g/h) (WAT vs. GLU, *p* < 0.001; GLU vs. TRE, *p* < 0.05).

## 4. Discussion

We compared the average power and maximum power values in each exercise set. We reported the data (Watts), and also expressed the baseline-normalized data (%), among the three trials (WAT, GLU, and TRE). In the final phase (the fourth set) of the test, both of the parameters (average power and maximum power) were significantly higher in the TRE trials than in the trials of the WAT and GLU, demonstrating the efficacy of TRE ingestion on Wingate-test performance (Figure 3a,b). In this exercise protocol, the blood glucose levels were increased after each set of the Wingate test in all three of the trials (Figure 5). During high-intensity exercise at more than 80% VO_2_max, the liver supplies GLU to the blood in an amount exceeding that taken up by skeletal muscles through the effect of catecholamines, resulting in a rapid increase in blood glucose levels [29,30,31,32,33,34]; the above-mentioned finding in our study is probably attributable to this phenomenon. However, the increase in blood glucose levels after the Wingate test decreased with the increasing number of sets in all three trials (⑥, ⑨, and ⑫ in Figure 4). When the liver glycogen decreases, the blood glucose levels are reportedly less likely to increase, despite an increase in the exercise intensity [35]; therefore, the above-mentioned decrease in blood glucose levels was presumed to reflect a gradual decrease in the liver glycogen. The blood glucose levels in the WAT trial remained to be significantly lower than those in the carbohydrate-ingestion trials (trials TRE and GLU), and tended to be lower than those at rest in the final phase of the examination (⑩, ⑪, and ⑫ in Figure 5). This finding reflects an obvious decrease in the liver glycogen in the WAT trial. The depletion of the liver glycogen was presumed to progress to a greater extent than that in the other two trials, preventing the glycolytic metabolism from producing the energy required for exercise performance.

At 30 min after test-drink ingestion, the blood glucose levels were significantly higher in the GLU trial than in the trials of TRE and WAT (⑤ in Figure 5). In the TRE trial, there was no rapid increase in the blood GLU levels, as observed in the GLU trial. In addition, in the final phase of the trial, the blood GLU levels in the TRE trial did not fall below those at rest, and were significantly higher than those in the WAT trial (⑤ to ⑫ in Figure 5). A preceding study compared the changes in the blood glucose levels during the 180-min rest period following the oral ingestion of 25 g of GLU and TRE. The blood glucose levels increased sharply 30 min after glucose ingestion, and then decreased rapidly. In contrast, the TRE induced no rapid increase in the blood glucose levels, and the blood glucose levels at 120 and 180 min after ingestion were higher for TRE than for GLU [24]. In our study, the GLU trial exhibited a rapid increase in the blood glucose levels after the ingestion of a GLU solution. This phenomenon was presumed to enhance the glucose metabolism (Figure 7) and to inhibit the lipid metabolism, resulting in a “waste of glucose”, meaning that a rapid increase in the blood glucose may lead to the rapid use of glucose, and it eventually resulted in the near-complete depletion of glucose storage, even in a low intensity prolonged exercise. In the TRE trial, no rapid increase in the blood GLU levels occurred, thus, we speculate that it may lead to a slight insulin response without enhancing the glucose metabolism; this might further lead to the use of more lipids for energy production, and to the preservation of carbohydrates. In Part 2 of the current study, we examined our speculations—the effective use of lipid by TRE ingestion during prolonged exercise.

Energy production by glycolytic metabolism is required to a greater extent in exercise, as determined by the average power value, than in that by the maximum power value. In our study, the average power value was significantly higher in the carbohydrate-ingestion trials (TRE and GLU) than in the WAT trial from the third set. In the fourth set, the parameter was significantly higher in the TRE trial than in the GLU and WAT trials.

The maximum power value, which is obtained during the first 5 to 10 s of each bout of exercise, energy produced by the ATP-Pcr pathway accounts for more than half of the total energy required, whereas that produced by the glycolytic pathway accounts for about 30% [36]; therefore, this type of exercise (i.e., maximum power) may be less influenced by the depletion of carbohydrates, as compared with that, as determined by the average power value. In our study, the maximum power value was significantly higher in the TRE trial than in the trials of GLU and WAT in the fourth set.

Therefore, a significant inter-trial difference in performance, due to the depletion of carbohydrates (muscle and liver glycogen), occurred earlier in the analysis results of the average power value. These circumstances could explain the differences in the performance between the analysis results of the two parameters (average power value and maximum power value). In both parameters, the performance tended to decrease with the increasing number of sets in all of the trials, indicating that the exercise performance was influenced by the depletion of the carbohydrates. However, the maximum power value did not decrease in the TRE trial, suggesting that the depletion of carbohydrates did not progress to such an extent that it affects the performance, and thus that the carbohydrates were preserved.

After the end of the fourth set, the blood lactate levels were higher in the TRE trial than in the other two trials (⑫ in Figure 6). This was also believed to be because more glucose could be degraded in the later phase of the prolonged exercise in the TRE trial than in the other two trials, supporting the preservation of the total glucose in exercising men [37,38]. Regarding the effects of energy depletion on performance, as shown in the analysis results of the RER (Figure 6), the level of glucose metabolism remained higher in the GLU trial than in the other two trials after the ingestion of a glucose solution; this was presumed to cause the depletion of muscle glycogen, to be progressed to a higher extent in the GLU trail than in the TRE trial, resulting in a difference in performance between these trials in the third set for the average power value, and fourth set for the maximum power value. As discussed above, the TRE is absorbed slowly and gradually increases the blood GLU levels and continuously supplies glucose to the blood when ingested during exercise, consisting of ultra-high-intensity intermittent exercise and constant-load exercise, showing that TRE is effective in maintaining or improving performance in such exercise.

In Part 2, we assessed the effects of a single ingestion of a TRE solution on energy metabolism during low-intensity constant-load exercise requiring fatty acid burning to a higher extent. After 15 min of the test-drink ingestion (③ in Figure 8), the blood glucose levels were increased mildly for the TRE trial, sharply for the GLU trial, and decreased slightly for the WAT trial—the three trials exhibited significantly different blood glucose levels from each other because of the effects of the carbohydrate ingestion. This result may reflect the properties of TRE, which is absorbed slowly and gradually increases the blood glucose levels, compared with GLU, which rapidly increases the blood glucose levels immediately after being ingested. The decrease in the blood glucose levels in the trial WAT is probably attributable to the near-depletion of the liver glycogen.

The lower the RER, the more lipids being burned (lipid oxidation) [25]. The analysis results of the RER (Figure 9) appear to reflect the enhancement of the glucose metabolism in the two carbohydrate-ingestion trials, compared with the WAT trial. In addition, the RER remained significantly lower in the TRE trial than in the GLU trial from 30 min after carbohydrate ingestion to the end of the examination, which may also reflect the properties of TRE. Trehalose takes a longer time to be absorbed, because of requiring degradation by TRE; this may explain why, in the TRE trial, the glucose metabolism was not enhanced as much as in the GLU trail, resulting in lower RER levels during that period of time. As in the preceding studies [23,24], the TRE ingestion also appeared to induce only a slight insulin response in this experiment. Therefore, after the TRE ingestion, the insulin might not promote lipid synthesis as much as after the glucose ingestion, and thus as many lipids could be burned in order to continue to exercise. This was believed to explain why the total lipid oxidation was significantly higher in the TRE trial than in the GLU trial (Figure 9).

This study has some strengths. We recruited sufficient numbers of participants to investigate the exercise performances and energy metabolism of the TRE ingestion among healthy young adults. To the best of our knowledge, our study is one of the first studies where researchers focused on both the exercise performances and energy metabolism of a single ingestion of TRE during prolonged exercise. However, our results require confirmation in a larger study, which should determine the mechanisms of the effects of the TRE we observed. This study has several limitations. First of all, we decided to have a convenience sampling only including male participants, which does not allow for us to generalize our results to the whole population of healthy young adults. Second, we did not ask the participants to keep a food diary, and did not place any restriction on the diet, including caffeine consumption. We do not know whether the participants had a high-carbohydrate diet or habitual caffeine consumption [39], and thus the results of our study may have been affected. It is possible that standardizing the diets of the participants may have enhanced our ability to detect more effects of the TRE intervention. Third, the use of capillary blood sampling techniques limited the number of biomarkers that we could measure. The collection of larger blood samples would have enabled us to measure more comprehensive hormonal changes, as well as a range of markers of energy metabolism and oxidative stress.

As discussed above, TRE was suggested to inhibit the enhancement of glucose metabolism and increase the use of lipids, thereby allowing for the preservation of carbohydrates during prolonged exercise. Although central fatigue was not determined in this experiment, preventing hypoglycemia may have a certain effect, not only on the maintenance of performance, but also on glucose metabolism in the brain [40,41], and thus could also contribute to the reduction of central fatigue.

## 5. Conclusions

In the case where only a single energy supplementation is available during prolonged exercise comprised of ultra-high-intensity intermittent exercise, TRE ingestion was shown to be more effective in maintaining or improving exercise performance than GLU ingestion. This may be because TRE enabled the effective use of lipids, leading to the preservation of energy until the later stages of exercise. On the basis of the outcomes of our study, TRE could potentially be applied as a functional sports drink in the future.

## Figures and Tables

**Figure 1 sports-07-00100-f001:**
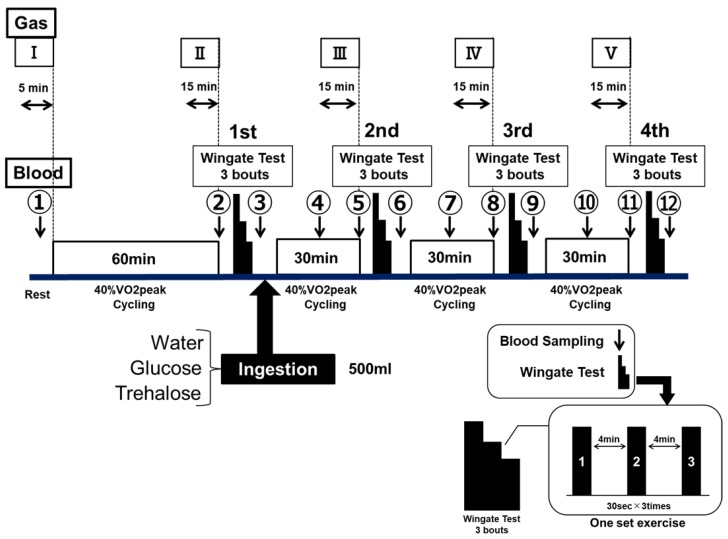
Experimental design—Part 1.

**Figure 2 sports-07-00100-f002:**
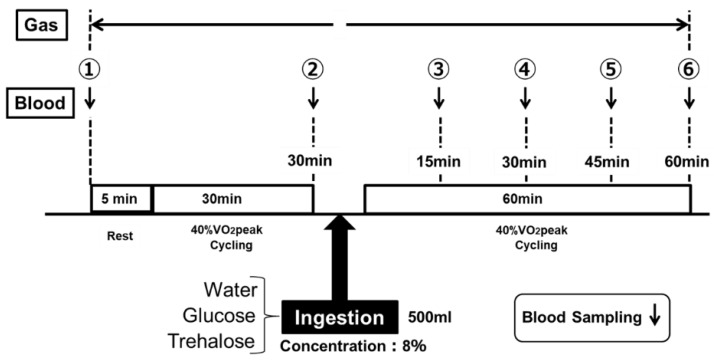
Experimental design—Part 2.

**Figure 3 sports-07-00100-f003:**
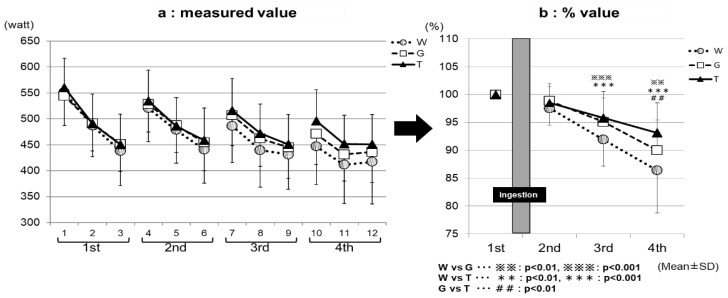
The average power assessments. * W—water; G—glucose; T—trehalose. Comparison of the average power value among the exercises (1–4 sets) for each trial (water (WAT), glucose (GLU), or trehalose (TRE)). (**a**) The measured value. (**b**) For each trial, the values for the Wingate-test performance in the second to fourth set were compared with and expressed as percentages of that in the first set.

**Figure 4 sports-07-00100-f004:**
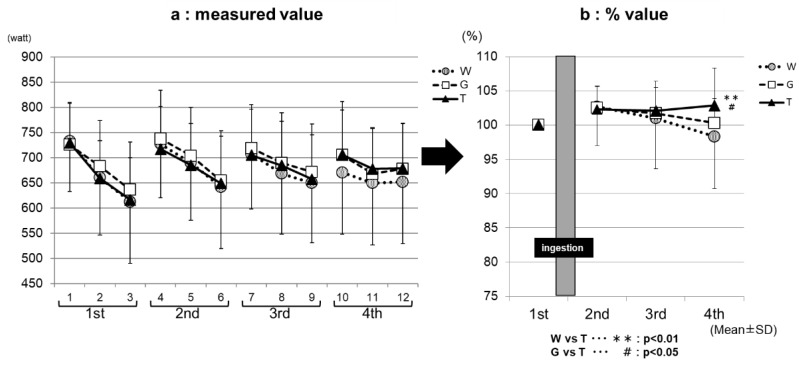
The maximum power assessments. * W—water; G—glucose; T—trehalose. Comparison of the maximum power value among the exercises (1–4 sets) for each trial (WAT, GLU, or TRE). (**a**) The measured value. (**b**) For each trial, the values for the Wingate-test performance in the second to fourth set were compared with and expressed as percentages of that in the first set.

**Figure 5 sports-07-00100-f005:**
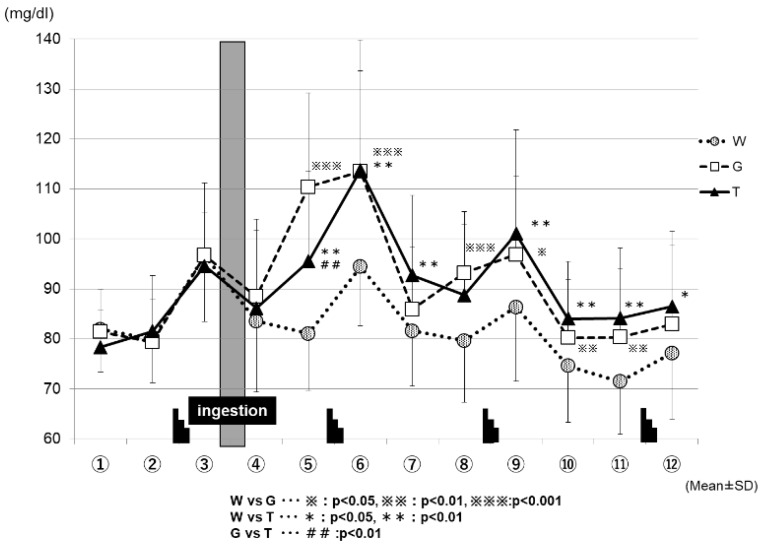
Changes in the blood glucose levels for each trial. * W, water; G, Glucose; T, Trehalose.

**Figure 6 sports-07-00100-f006:**
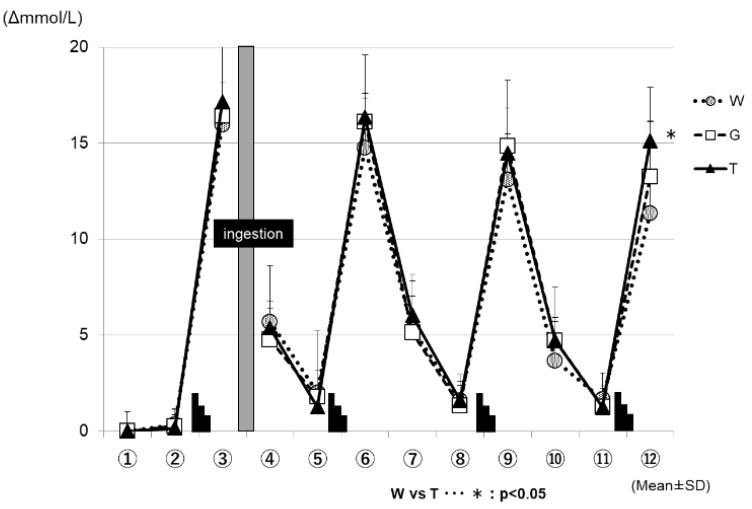
Changes in the blood lactate levels (Δ) for each trial. * W—water; G—glucose; T—trehalose.

**Figure 7 sports-07-00100-f007:**
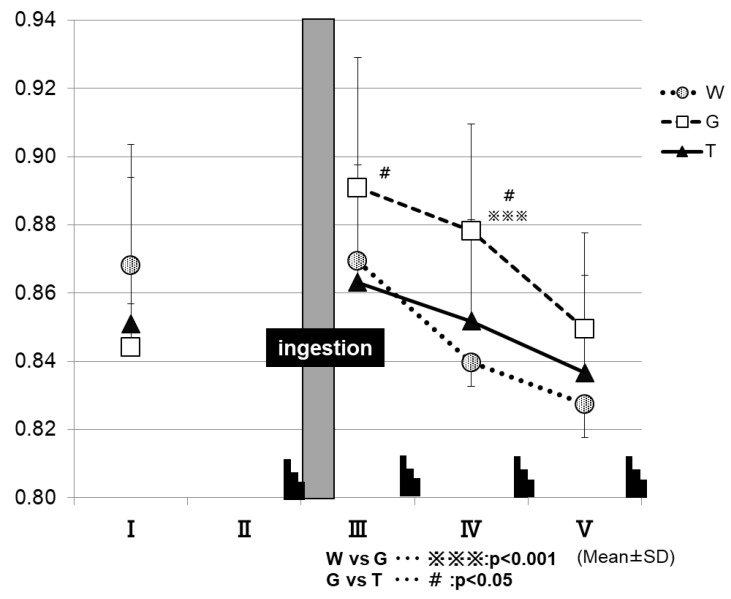
Changes in respiratory exchange ratio (RER) for each trial. *W—water; G—glucose; T—trehalose.

**Figure 8 sports-07-00100-f008:**
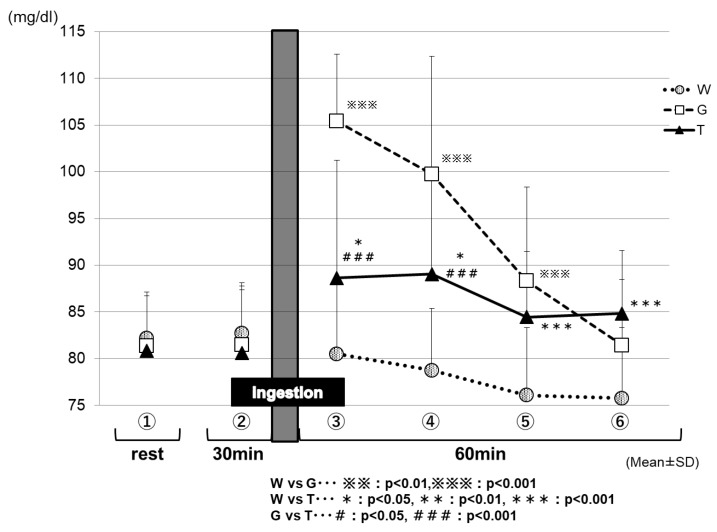
Comparison of the blood glucose levels for each trial. After the ingestion of the test drink, each trial exhibited changes in the blood glucose levels, reflecting the properties of its test drink. * W—water; G—glucose; T—trehalose.

**Figure 9 sports-07-00100-f009:**
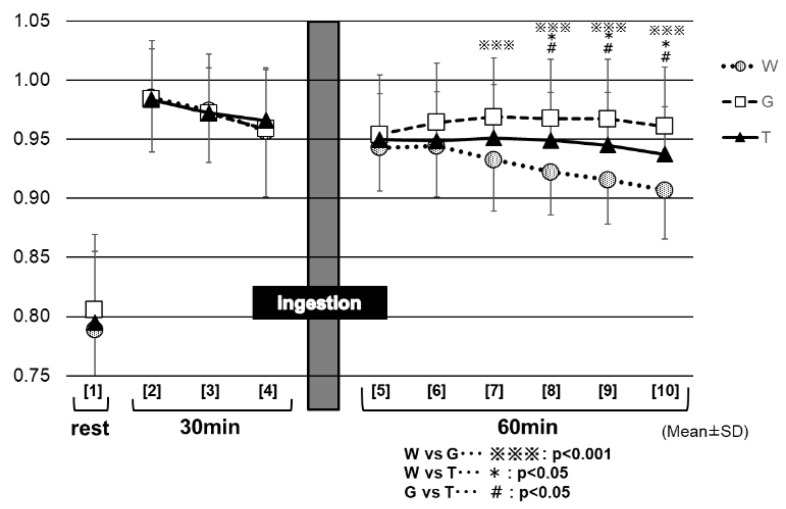
RER for each trial. * W—water; G—glucose; T—trehalose.

**Figure 10 sports-07-00100-f010:**
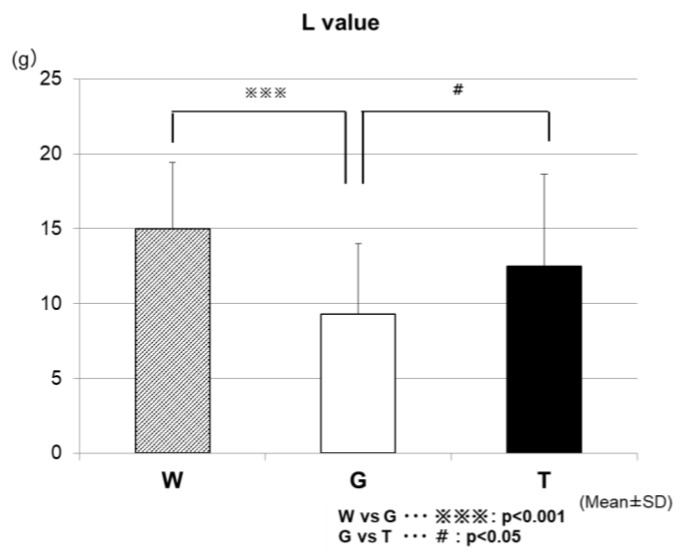
Comparison of the total lipid oxidation (L-value) during the 60-min exercise among the three trials. * W—water; G—glucose; T—trehalose.

**Table 1 sports-07-00100-t001:** Physical characteristics of the participants (Part 1). SD—standard deviation.

n = 25	Age (year)	Height (cm)	Weight (kg)	% fat (%)	VO_2_ (peak) (mL/min)	Load (peak) (watt)	40%VO_2_peak (mL/min)	Load 40%VO_2_peak (watt)
Mean	21.3	172.7	66.4	19.2	2812.4	251.7	1125.0	97.7
SD	1.1	5.0	6.9	2.5	380.6	33.1	152.3	14.6

**Table 2 sports-07-00100-t002:** Physical characteristics of the participants (Part 2).

n = 24	Age (year)	Height (cm)	Weight (kg)	%fat (%)	VO_2_ (peak) (mL/min)	Load (peak) (watt)	40%VO_2_peak (mL/min)	Load 40%VO_2_peak (watt)
Mean	21.9	173.4	66.2	19.6	2988.5	253.7	1195.4	99.0
SD	1.2	5.6	5.8	2.8	368.1	32.7	147.2	15.7

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
