# Peer review of "Effects of a Single Ingestion of Trehalose during Prolonged Exercise"

_sports, 2019, doi:10.3390/sports7050100_

Round 1
Reviewer 1 Report
General Thoughts:
Overall, the manuscript is well written. However, I am not sure on the rational of doing comparing two different studies. It appears that you are comparing two different modes to each other but I could be interpreting it in correctly. I suggest that the authors go back to the drawing board and if it is truly two different studies then they just to choose either study 1 or study 2.
Methods:
Line 87: What was the training status of your participants?
Line 98: Did any of the participants take caffeine regular.
Line 132: How was V02peak defined?
Line 146: Was the measurements of glucose and lactate taken in duplicates?
Results: Why was RER examined for study 1? I thought the aim of this study was to examine single ingestion of trehalose on exercise performance.
Discussion: Overall, well written. Just needs to discuss either study 1 or study 2.
Author Response
First, we would like to thank the reviewers for their comments and recommendations. We appreciate the time and effort it takes to review a manuscript and make thoughtful comments and suggestions. We have responded to these comments and listed our point-by-point responses to the reviewers below.
Reviewer 1
Overall, the manuscript is well written. However, I am not sure on the rational of doing comparing two different studies. It appears that you are comparing two different modes to each other but I could be interpreting it in correctly. I suggest that the authors go back to the drawing board and if it is truly two different studies then they just to choose either study 1 or study 2.
o First, we thank you for your comments. As you commented, our explanation of the entire study was not clear. The current study was not comparing two different studies. Actually, it was one study examining the effectiveness of trehalose ingestion of prolonged exercise due to enhancing lipid oxidation. Thus, the aim of this study was twofold. First, we aimed to examine the exercise performance comparing the glucose and trehalose. The second purpose was to investigate the benefits of trehalose regarding its energy metabolism.
o We observed the changes in respiratory exchange ratio (RER) in the first part of the study, examining the benefits of trehalose on exercise performance (in Study 1). The result of RER confirmed that trehalose was acting differently during prolonged exercise in terms of energy metabolism compared to glucose ingestion. (Figure 6) In the following assessments of energy metabolism, we confirmed our hypothesis that a single ingestion of trehalose solution would promote a greater performance benefits due to higher lipid oxidation and maintaining higher blood glucose levels in trehalose trial (remaining RER to be significantly lower) than glucose trial.
o Since this is one study, we revised the entire manuscript replacing “Study 1” and “Study 2” to “Part 1” and “Part 2”, responding your comments and confusion of “comparing two different studies”, “two different modes….”
o We revised the entire manuscript using “Part 1- exercise performance” and “Part 2 - energy metabolism”. We think this is appropriate wording and help readers understand our study is one study, not two separate studies. This is very important, and again, we greatly appreciate your comments.
Methods:
Line 87: What was the training status of your participants?
o Our participants were not student athletes who train for competing in their sports. They all were “recreationally trained” college students. We added “recreationally trained” in the sentence, as Healthy recreationally trained college male students (Line 109)
o We also described about participants training status under Methods, “Participants were also instructed to maintain their normal diet and physical activity during the examination period. In addition, they were not allowed to perform vigorous physical activity, ingest caffeine, or drink alcohol from 24 hours before each examination….” (Lines 120 – 123)
o We do not have the information about the participants’ training or physical activity status – we will add this as our study limitation.
Line 98: Did any of the participants take caffeine regular.
o We do not have the information about the participants’ regular intake of caffeine. We added this as our study limitation.
Line 132: How was V02peak defined?
o Definition of VO2peak, the highest value of VO2 attained during maximum physical effort, designed to bring the participant to the limit of tolerance, was added. (Line 138-139)
Line 146: Was the measurements of glucose and lactate taken in duplicates?
o Yes. We measured the blood glucose and lactate in duplicates. When the differences of the measurements were more than 10%, we took the third measurement. We added this procedure in Method section (Line 169)
Results: Why was RER examined for study 1? I thought the aim of this study was to examine single ingestion of trehalose on exercise performance.
o RER, indicator of substrate use in tissues, was measured after ingestion of test solutions in Study 1 (Part 1 - exercise performance). As we expected, after ingestion of glucose RER was significantly higher in glucose trial than in trehalose trial indicating enhancing glucose metabolism in glucose trial, but not in trehalose trial at the time points of III to IV (but not at the time point of V) in Part 1. We also examined the changes in blood glucose and lactate in both glucose and trehalose trials in Study 1 (Part1). There were significant differences in blood glucose and lactate in glucose and trehalose trials during the last phase of testing (time point of V). These were very interesting findings that made us design the Study 2 (Part2) to examine substrate utilization in prolonged exercise without breaks (having Wingate tests) as in Study 1 (Part1).
o In summary, after we have observed the changes of RER, blood glucose and lactate in Study 1 (Part 1), we examined to see whether trehalose ingestion enhance lipid oxidation during the prolonged exercise (continuation of 60 min cycling at 40% VO2 peak) or not. Our findings in Study 2 (Part 2) provide novel findings about trehalose ingestion for endurance athletes, such as marathon runners.
Discussion: Overall, well written. Just needs to discuss either study 1 or study 2.
o Please see above our responses. We revised Discussion section emphasizing the reasons why we designed Study 1 and Study 2 (Part 1 and Part 2).
Reviewer 2 Report
The article entitled “Effects of a single ingestion of trehalose during prolonged exercise” under consideration for publication in Sports, compared the effect of trehalose versus glucose or water for effect on athletic performance. In general, the study seems well-designed (although trial 1 seems very difficult for subjects to complete), however there are some points that require clarification (listed below).
Major Considerations
In general, the manuscript could benefit from a thorough English proofreading. For example, the use of “to be” is unnecessary in many cases (lines 209, 260, 293, 335, 336, 358, 370). Other examples of typographical corrections are listed below under minor considerations.
The authors should clarify in the abstract that two distinct studies were performed to clarify the differences in subject composition between the different experiments.
This reviewer wants to confirm that they are understanding the methods correctly. In experiment #1, subjects completed over 3 hours of continuous exercise including 12 maximum effort wingate protocols? This seems very difficult for subjects to complete. The authors should provide a clear rationale for this procedure.
Timing of gas measurements should be defined in Figure 2.
Lines 141-143: This sentence is unclear and should reworded. The authors should state exactly how average power over the 3 wingate clusters were calculated.
In this reviewer’s mind, Figure 3 (both panels) would benefit from the presentation of untransformed data alongside the baseline-normalized data. This seems important given subjects only experienced subtle decreases in average power (and almost no decrement in peak power) following a very difficult exercise protocol.
Statistical Methods: The authors should state what correction factor was used for multiple comparisons at each time point.
Line 310: the authors suggest that glucose is wasted, but that does not seem like an accurate description. Please reword for accuracy.
Line 310-311: the authors state that only a slight insulin response occurred, yet insulin levels were not measured or reported. The authors should state that this is speculation.
Discussion Section: Was diet controlled for beyond the restrictions listed in line 98? If not, this should be listed as a limitation of the study in the Discussion.
Minor Considerations
Line 42: The authors state “In this method, the activity of glycogen synthase is enhanced by reducing muscle glycogen in advance, thereby storing more glycogen than usual in the body [10].” Please elaborate on this mechanism, because as written, it implies simply deplete glycogen increases glycogen content (when in fact it is accomplished through increased carbohydrate consumption).
Line 49-50: this sentence doesn’t seem accurate and requires rewording.
Line 57: “For the reason” should read “For this reason”
Line 71-76: the authors state that trehalose causes no such increase in blood glucose, yet provide data in Fig 4 that trehalose increases blood glucose to a similar extent as glucose. The authors then go on to state that trehalose maintains a high blood glucose, for a longer period of time This seem conflicting and requires further clarification or rewording.
Line 83: “containing” should be “comprised of”
Line 223: “remained to be exhibited” should be “exhibited”
Line 316-319: this sentence is unclear and requires rewording.
Line 327: the word “definitely” should be removed.
Author Response
First, we would like to thank the reviewers for their comments and recommendations. We appreciate the time and effort it takes to review a manuscript and make thoughtful comments and suggestions. We have responded to these comments and listed our point-by-point responses to the reviewers below.
Reviewer 2
The article entitled “Effects of a single ingestion of trehalose during prolonged exercise” under consideration for publication in Sports, compared the effect of trehalose versus glucose or water for effect on athletic performance. In general, the study seems well-designed (although trial 1 seems very difficult for subjects to complete), however there are some points that require clarification (listed below).
We appreciate your great comments and detailed review for our manuscript. We revised the entire manuscript including all your recommendations. First, we would like to let you know about the participants in this study.
Trial 1 – Study 1, exercise performance assessment protocol (Figure 1) was difficult for the participants, but our study participants completed all trials.
Study 2, energy metabolism assessment protocol (Figure 2) was performed by another group of recreationally trained college male students. They also completed all trials. There was no drop-outs in this study.
Major Considerations
In general, the manuscript could benefit from a thorough English proofreading. For example, the use of “to be” is unnecessary in many cases (lines 209, 260, 293, 335, 336, 358, 370). Other examples of typographical corrections are listed below under minor considerations.
o The original manuscript went through professional proofreading by native English speaker, but we checked all the lines (removed some of “to be”) and wording that you have mentioned here carefully. We appreciate your time and efforts for checking our manuscript!
The authors should clarify in the abstract that two distinct studies were performed to clarify the differences in subject composition between the different experiments.
o This is the great point about the study. As we responded Reviewer 1, we revised the entire manuscript emphasizing Study 1 and Study 2 are different, and distinct studies. Study 1, now we called it “Part 1” examined the exercise performance comparing the glucose and trehalose ingestions and Study 2, now we called it “Part 2” investigated the benefits of trehalose regarding its energy metabolism.
o As you have mentioned, subject composition between Study 1 and Study 2 were different. We have listed them in Tables 1 and 2.
This reviewer wants to confirm that they are understanding the methods correctly. In experiment #1, subjects completed over 3 hours of continuous exercise including 12 maximum effort wingate protocols? This seems very difficult for subjects to complete. The authors should provide a clear rationale for this procedure.
o This is another great point to confirm about the protocols in Study 1 (Part 1 – exercise performance). Experimental trial in Part 1 is difficult for young healthy recreationally trained adults to complete, but there was no drop-outs in our study. Everyone completed all trials.
o The rationale for this procedure was developing physiological stage of full marathon when runners start rely on fat for fuel, and start slowing down, hitting “runner’s wall.” (Added the rationale in Line – 154-156)
Timing of gas measurements should be defined in Figure 2.
o Timing of gas measurement in Study 2 (Part 2 – energy metabolism) was continuous. We explained the procedure of gas measurement in Part 2. (Line 177)
Lines 141-143: This sentence is unclear and should reworded. The authors should state exactly how average power over the 3 wingate clusters were calculated.
o In Method section, Assessment of Exercise Performance, we have explained how the Wingate test was done, in addition to your recommendation, how 3 Wingate clusters were calculated.
o In Part 1, exercise performance was assessed by using the Wingate test. The results of each Wingate test were expressed as 1) the average power values – Figure 3a and 2) the maximum power values – Figure 3b.
o The average power values (Watts) were determined for each 5-second period for 30 seconds, and they (5-seconds x 6 readings = 30 seconds) were averaged as a bout of Wingate test. The participants were allowed unloaded pedaling of 5 seconds to reach maximum cadence and they were instructed to maintain maximal pedal speed throughout the 30 seconds period.
o The maximum power value (Watts) was the highest power produced in a 30-second segment. The 30-second segment of the test were repeated 3 times between 4 minutes of resting period. This is one bout of Wingate test. (See Figure 1 - Wingate test) The participants were motivated verbally throughout the test.
In this reviewer’s mind, Figure 3 (both panels) would benefit from the presentation of untransformed data alongside the baseline-normalized data. This seems important given subjects only experienced subtle decreases in average power (and almost no decrement in peak power) following a very difficult exercise protocol.
o We have added the data without baseline-normalized data (Watts) alongside the original data expressed percentages of that in the first set as 100%. (See Figure 3)
Statistical Methods: The authors should state what correction factor was used for multiple comparisons at each time point.
o We have revised the section of Statistical Analysis. The obtained data are expressed as the mean ± standard deviations (SD). A two-way (factors: solution and time) ANOVA without replication was used for comparison of the data among the three trial groups. When a significant difference was detected, a multiple comparison test was performed using post hoc Bonferroni correction. All analyses were performed using SPSS version 25.0 (IBM Japan, Tokyo, Japan) and statistical significance was set at p < 0.05.
Line 310: the authors suggest that glucose is wasted, but that does not seem like an accurate description. Please reword for accuracy.
o We explained what we were trying to say in the statement using “waste of glucose” expression. “waste of glucose”, meaning that a rapid increase in blood glucose by GLU ingestion, that may lead to unnecessary rapid use of glucose and it is eventually resulted in the near-complete depletion of glucose storage even in low intensity prolonged exercise. (Line 342)
Line 310-311: the authors state that only a slight insulin response occurred, yet insulin levels were not measured or reported. The authors should state that this is speculation.
o This is another good point. We added “We speculated that it may lead to a slight insulin response ….” (Line 338)
Discussion Section: Was diet controlled for beyond the restrictions listed in line 98? If not, this should be listed as a limitation of the study in the Discussion.
o We did not control participants’ diets beyond the restrictions listed in instruction - to maintain their normal diet … during the examination period. In addition, they were not allowed to…., ingest caffeine, or drink alcohol from 24 hours before each examination and were fasted from 9 p.m. on the day before each examination.
o We have added not controlling their diet as limitations of the study. (Line 401)
Minor Considerations
Line 42: The authors state “In this method, the activity of glycogen synthase is enhanced by reducing muscle glycogen in advance, thereby storing more glycogen than usual in the body [10].” Please elaborate on this mechanism, because as written, it implies simply deplete glycogen increases glycogen content (when in fact it is accomplished through increased carbohydrate consumption).
o (Line 55-57) We have revised the mechanism of “glycogen loading” as follow, “In this method, muscle glycogen re-synthesis is enhanced by depleting the muscle glycogen stores by heavy physical activity and then eating a diet rich in carbohydrates (exhaustive depletion followed by a carbohydrate-rich diet).”
Line 49-50: this sentence doesn’t seem accurate and requires rewording.
o (Line 65-68) We have revised the sentence as follow, “However, because absorption properties and availability in skeletal muscles differ depending on the types of carbohydrates ingested before or during exercise [18, 19], TRE, paltinose (isomaltulose), etc. relatively new carbohydrates to sport nutrition, have not established methods of ingestions (i.e., timing, amount, etc.), which may enhance exercise performance.
Line 57: “For the reason” should read “For this reason”
o Changed “For the reason” to “For this reason” (Line 74)
Line 71-76: the authors state that trehalose causes no such increase in blood glucose, yet provide data in Fig 4 that trehalose increases blood glucose to a similar extent as glucose. The authors then go on to state that trehalose maintains a high blood glucose, for a longer period of time This seem conflicting and requires further clarification or rewording.
o We clarified these sentences for explaining the TRE functions in various conditions. We have revised the sentences as follow:
“It is reported that after ingestion of GLU or sucrose, lipid oxidation is inhibited by a rapid increase in blood GLU levels, whereas carbohydrate oxidation increases [25]. In contrast, as mentioned above, TRE causes no such rapid increase in blood GLU levels and induces only a slight insulin response. TRE takes longer to digest than most sugars, therefore, after ingestion of TRE, lipid oxidation may be less likely to be inhibited, leading to preservation of carbohydrates; this indicates that TRE is suitable as a carbohydrate to be ingested before prolonged exercise. Utilization of such physiological properties of TRE could maintain higher blood GLU levels than ingestion of GLU for a longer period of time, leading to maintenance or improvement of performance during the later stages of prolonged exercise.
Line 83: “containing” should be “comprised of”
o Changed “containing” to “comprised of” (Line 418)
Line 223: “remained to be exhibited” should be “exhibited”
o Changed “remained to be exhibited” to “exhibited” (Line 254)
Line 316-319: this sentence is unclear and requires rewording.
o Actually, part of this sentence was redundancy of the next sentence. We have revised the sentences as follow:
“In Wingate-test exercise as determined by the maximum power value, which is obtained during the first 5 to 10 seconds of each bout of exercise, energy produced by the ATP-Pcr pathway accounts for more than half of the total energy required, whereas that produced by the glycolytic pathway accounts for about 30% [36]; therefore, this type of exercise (i.e., maximum power) may be less influenced by the depletion of carbohydrates, as compared with that as determined by the average power value. Therefore, a significant inter-trial difference in performance due to the depletion of carbohydrates (muscle and liver glycogen) occurred earlier in the analysis results of the average power value.”
Line 327: the word “definitely” should be removed.
o Removed “definitely” (Line 357)
Reviewer 3 Report
The study that is shown is very interesting and novel, however, I think it is very presumptuous for the authors to indicate this topic in the conclusions of the abstract. The ideal if you want to indicate it is to put something like "for the knowledge of the authors ....". However, the abstract should be reviewed in general.
On the other hand, the inclusion of 10 women in study 2 without comparing the effects with men could be a limitation. Women due to the menstrual cycle are subject to hormonal changes that can affect performance. In addition, it is not indicated how these women have been randomized. How many women have been in each group? Perhaps they should be eliminated from the study or include the gender as a confounding variable in the statistical analysis.
Finally, it would be very interesting that the authors indicate some type of limitation and strength that the study has.
Author Response
First, we would like to thank the reviewers for their comments and recommendations. We appreciate the time and effort it takes to review a manuscript and make thoughtful comments and suggestions. We have responded to these comments and listed our point-by-point responses to the reviewers below.
Reviewer 3
The study that is shown is very interesting and novel, however, I think it is very presumptuous for the authors to indicate this topic in the conclusions of the abstract. The ideal if you want to indicate it is to put something like "for the knowledge of the authors ....". However, the abstract should be reviewed in general.
o We have revised abstract significantly based on the comments provided by you, and reviewers 1 and 2. Thank you!
On the other hand, the inclusion of 10 women in study 2 without comparing the effects with men could be a limitation. Women due to the menstrual cycle are subject to hormonal changes that can affect performance. In addition, it is not indicated how these women have been randomized. How many women have been in each group? Perhaps they should be eliminated from the study or include the gender as a confounding variable in the statistical analysis.
o Great comment on gender differences. We have eliminated women from our study, recalculated the data and presented the study only with male participants.
Finally, it would be very interesting that the authors indicate some type of limitation and strength that the study has.
o We have added limitations and strength of the current study. (Line 394)
o This study has some strengths. We recruited sufficient numbers of participants to investigate exercise performances and energy metabolism of trehalose ingestion among healthy young adults. To the best of our knowledge, our study is one of the first study that researchers focused on both exercise performances and energy metabolism on a single ingestion of trehalose during prolonged exercise. However, our results require confirmation in a larger study, which should determine the mechanisms of effects of TRE we observed.
o This study has several limitations. First of all, we decided to have a convenience sampling only including male participants, which does not allow us to generalize our results to the whole population of healthy young adults. Second, we did not ask participants to have a food diary and did not place any restriction on the diet, including caffeine consumption of our participants. If participants have a high-carbohydrate diet or habitual caffeine consumption [Germaine et al, Sports, 2019], the results of our study may have been affected. It is possible that standardizing the diets of the participants may have enhanced our ability to detect more effects of the TRE intervention. Third, the use of capillary blood sampling techniques limited the number of biomarkers that we could measure. The collection of larger blood samples would have enabled us to measure a more comprehensive hormonal changes and a range of markers of energy metabolism and oxidative stress.
Round 2
Reviewer 1 Report
Overall,
The authors have addressed my concerns sufficiently.
Author Response
We appreciate your time and efforts for reviewing our revision.
Reviewer 2 Report
In general, the revised report by Wadazumi et al. entitled “Effects of a single ingestion of trehalose during prolonged exercise” under consideration for publication in Sports, is much improved, however the following considerations remain.
Major Considerations
In general, there are still numerous typographical and linguistic issues throughout (especially within the background). Some of the issues are itemized below, however, the article should undergo another proofreading.
Line 37: The authors state: “Glycogen stored in the body alone leads to energy deficiency during the later stages of the prolonged exercise [4].” This is not accurate. Please reword or clarify what you are trying to communicate.
Line 53: The authors state: “TRE, paltinose (isomaltulose), etc., relatively new carbohydrate to sport nutrition, have not established methods of ingestions (i.e., timing, amount, etc.), which may enhance exercise performance.” This is unclear and should be reworded. I believe the authors are trying to say that foundational data regarding nutrient dosing and timing has yet to be established for TRE.
Line 86: “GLU” should not be used as short hand for glucose when referring to blood glucose as the study also contains a group abbreviated “GLU” (possibly causing confusion as is). This occurs in line 71, 74, 75, 86, 224, 225 (twice), 228, 232, 234, 237, 264, 266, 268, 271, 274, 275, 308, 311, 316, 325 (twice), 327, 328, 330 (twice), 334, 336, 367, 373, 375, 377, 378. GLU should only refer to the treatment condition. The figures (and legends) should also be updated to reflect these new group designations for clarity and continuity between the figures and manuscript text.
Minor Considerations
Line 16: “First, we aimed to examine the exercise performance comparing the ingestions of GLU and TRE.” should read “First, we examined exercise performance following ingestion of either GLU, TRE, or WAT.”
Line 17: “investigate the benefits of TRE regarding its energy metabolism during” should be “investigate the effects of TRE on energy metabolism during”
Line 29: “GLU Lipid” should read “GLU. Lipid”
Line 36: “of the game” is unclear because the authors haven’t defined a “game” so “exercise” or “physical activity” are better descriptors.
Line 64: delete “are”
Line 81 The authors state: “However, only few studies have investigated the relationship between TRE ingestion and exercise performance from such a perspective [26].” but only cite one paper and without any detail. This study seems rather important and should be expanded upon.
Line 95: should specifically state that Part 2 had a different cohort of subjects than Part 1.
Line 343: “Energy 343 production by glycolytic metabolism is required to a greater extent in exercise as determined by the 344 average power value than in that by the maximum power value.” Through 352.
The misuse of “to be” is still present on several occasions in lines 85, 237, 283, 317, 363, 382. Please revise.
Author Response
Reviewer 2
Comments and Suggestions for Authors
In general, the revised report by Wadazumi et al. entitled “Effects of a single ingestion of trehalose during prolonged exercise” under consideration for publication in Sports, is much improved, however the following considerations remain.
Major Considerations
In general, there are still numerous typographical and linguistic issues throughout (especially within the background). Some of the issues are itemized below, however, the article should undergo another proofreading.
o We have used a professional English editing service (by native English speaking person) for this manuscript. However, we did careful proofreading for this second revision, again.
Line 37: The authors state: “Glycogen stored in the body alone leads to energy deficiency during the later stages of the prolonged exercise [4].” This is not accurate. Please reword or clarify what you are trying to communicate.
o We have revised the sentence. “The later stages of the prolonged exercise cause dramatic reductions in muscle glycogen content in skeletal muscle.”
Line 53: The authors state: “TRE, paltinose (isomaltulose), etc., relatively new carbohydrate to sport nutrition, have not established methods of ingestions (i.e., timing, amount, etc.), which may enhance exercise performance.” This is unclear and should be reworded. I believe the authors are trying to say that foundational data regarding nutrient dosing and timing has yet to be established for TRE.
o We have revised the sentence including your suggestions. “Foundational data regarding nutrient dosing and timing, which may enhance exercise performance has yet to be established for TRE.”
Line 86: “GLU” should not be used as short hand for glucose when referring to blood glucose as the study also contains a group abbreviated “GLU” (possibly causing confusion as is). This occurs in line 71, 74, 75, 86, 224, 225 (twice), 228, 232, 234, 237, 264, 266, 268, 271, 274, 275, 308, 311, 316, 325 (twice), 327, 328, 330 (twice), 334, 336, 367, 373, 375, 377, 378. GLU should only refer to the treatment condition. The figures (and legends) should also be updated to reflect these new group designations for clarity and continuity between the figures and manuscript text.
o We changed GLU to glucose for blood glucose. We used GLU for only treatment condition in the text. But in the abstract, we kept GLU.
o In Figures we have tried to use GLU, but that affects the images of Figures. Thus, we chose G for glucose, and mentioned in all the Figures.
Minor Considerations
Line 16: “First, we aimed to examine the exercise performance comparing the ingestions of GLU and TRE.” should read “First, we examined exercise performance following ingestion of either GLU, TRE, or WAT.”
o Revised as your recommended.
Line 17: “investigate the benefits of TRE regarding its energy metabolism during” should be “investigate the effects of TRE on energy metabolism during”
o Revised as you recommended.
Line 29: “GLU Lipid” should read “GLU. Lipid”
o Added period.
Line 36: “of the game” is unclear because the authors haven’t defined a “game” so “exercise” or “physical activity” are better descriptors.
o We have changed the word, “game” to “exercise’.
Line 64: delete “are”
o Deleted.
Line 81 The authors state: “However, only few studies have investigated the relationship between TRE ingestion and exercise performance from such a perspective [26].” but only cite one paper and without any detail. This study seems rather important and should be expanded upon.
o We added some detail about the study as an example. “Jentjens and Jeukendrup [26] determined the effects of pre-exercise ingestion of TRE on metabolic responses at rest and during exercise and on subsequent time-trial performance. They reported the ingestion of TRE leads to lower glucose and insulin responses prior to exercise and reduces the prevalence of rebound hypoglycaemia compared with the ingestion of glucose. However, TRE ingested before exercise did not affect time-trial performance.
Line 95: should specifically state that Part 2 had a different cohort of subjects than Part 1.
o We stated that Part 1 and Part 2 are different. “Any of the participants in Group 1 were not included in Part 2 experiment.”
Line 343: “Energy 343 production by glycolytic metabolism is required to a greater extent in exercise as determined by the 344 average power value than in that by the maximum power value.” Through 352.
o We have re-structure the paragraph to explain the differences of average power value and the maximum power value. (line 352-371)
The misuse of “to be” is still present on several occasions in lines 85, 237, 283, 317, 363, 382. Please revise.
o Revised the all these lines with “to be.”
Submission Date
30 March 2019
Date of this review
24 Apr 2019 20:04:38
Reviewer 3 Report
The authors have made a great effort to improve the manuscript. Now I think that if the editor considers it could be accepted.
Author Response

(The authors gave the same response as above.)
